# The Performance of Post-Fall Detection Using the Cross-Dataset: Feature Vectors, Classifiers and Processing Conditions

**DOI:** 10.3390/s21144638

**Published:** 2021-07-06

**Authors:** Bummo Koo, Jongman Kim, Yejin Nam, Youngho Kim

**Affiliations:** Department of Biomedical Engineering, Yonsei University, Wonju 26493, Korea; bmk726@ybrl.yonsei.ac.kr (B.K.); jmkim0127@ybrl.yonsei.ac.kr (J.K.); namyj1007@ybrl.yonsei.ac.kr (Y.N.)

**Keywords:** fall detection, artificial neural network, support vector machine, cross-dataset

## Abstract

In this study, algorithms to detect post-falls were evaluated using the cross-dataset according to feature vectors (time-series and discrete data), classifiers (ANN and SVM), and four different processing conditions (normalization, equalization, increase in the number of training data, and additional training with external data). Three-axis acceleration and angular velocity data were obtained from 30 healthy male subjects by attaching an IMU to the middle of the left and right anterior superior iliac spines (ASIS). Internal and external tests were performed using our lab dataset and SisFall public dataset, respectively. The results showed that ANN and SVM were suitable for the time-series and discrete data, respectively. The classification performance generally decreased, and thus, specific feature vectors from the raw data were necessary when untrained motions were tested using a public dataset. Normalization made SVM and ANN more and less effective, respectively. Equalization increased the sensitivity, even though it did not improve the overall performance. The increase in the number of training data also improved the classification performance. Machine learning was vulnerable to untrained motions, and data of various movements were needed for the training.

## 1. Introductions

Falls are one of the leading causes of death among the elderly [1]. Approximately 28–38% of people over 65 suffer a fall each year [2]. Falls can result in bruises and swellings, as well as fractures and traumas [3]. In addition to the physical consequences, the fear of falling can impact on the elderly’s quality of life. A fear of falling is associated with a decline in physical and mental health, and an increased risk of falling [4]. Therefore, falls and fall-related injuries are major healthcare challenges to overcome.

Many studies have tried to improve the physical performance of the elderly by performing rehabilitation programs to help prevent falls. Røyset et al. [5] conducted a fall prevention program using the Norwegian version of the fall risk assessment method, “STRATIRY” (score 0–5), but achieved no significant improvement when compared to the control group during a short stay in an orthopedic department. Gürler et al. [6] proposed a recurrent fall prevention program including assessment of fall risk factors, education on falls and home modification. This program was effective in reducing fall-related risk factors and increasing fall knowledge. Palestra et al. [7] presented a rehabilitation system based on a customizable exergame protocol (KINOPTIM) to prevent falls in the elderly. As a result of training for 6 months, the performance of the postural response was improved by an average of 80%.

Prevention of falls through long-term rehabilitation programs is important to improve the quality of life for the elderly, but preparation for the situation of a fall is also important. Falls may have serious consequences; however, most of the consequences of these falls are not directly attributed to the falls themselves, but to the lack of timely assistance and treatment [8]. Vellas et al. [9] reported that 70% of older adults who had fallen at home were unable to get up unaided, and that more than 20% of patients admitted to hospital as a result of a fall had been on the ground for an hour or more. Moreover, 50% of people affected by falls who remain unassisted for more than an hour die within the subsequent six-month period following the accident [10]. A fall-detection algorithm that detects and notifies the occurrences of falls as quickly as possible is important.

In general, an inertial measurement unit (IMU) sensor has been used for fall detection. Threshold-based methods were mainly used for fall detection [11,12,13]. They are advantageous because of a smaller computational time and can detect falls before the impact occurs. However, rapid movements can sometimes be misrecognized as falls. The machine learning-based algorithms require a relatively long computational time, but can distinguish similar actions accurately. It has been reported that machine learning-based algorithms perform better than threshold-based algorithms [14]. For this reason, threshold-based algorithms are mainly used to protect using wearable airbags through pre-impact fall detection [15], and machine learning-based algorithms are used to detect post-falls.

Several classifiers, such as support vector machine (SVM), k-nearest neighbor (k-NN), naïve Bayes (NB), least square method (LSM), artificial neural network (ANN) and others are used to detect post-falls. Researchers attempted to compare the classifiers which were suitable for post-fall detection [16,17,18]. Vallabh et al. [16] used smartphones placed in the trouser pockets to distinguish seven activities of daily living (ADLs) and four falls. They extracted data within the interval of acceleration between –20 m/s^2^ and 20 m/s^2^, and extracted feature vectors, such as the mean, median and skewness from this interval. Five classifiers were compared: LSM (75.4%) < NB (80.0%) < ANN (85.9%) < SVM (86.8%) < k-NN (87.5%). Özdemir et al. [17] used six IMU sensors to distinguish 16 ADLs and 20 falls. They extracted data within the 2 s window based on the impact, and extracted feature vectors, such as the mean, variance, skewness, kurtosis and so on from this interval. Six classifiers were compared: ANN (95.68%) < dynamic time warping (DTW) (97.85) < Bayesian decision making (BDM) (99.26%) < SVM (99.48%) < LSM (99.65%) < k-NN (99.91%). Gibson et al. [18] used one IMU sensor on the chest. They extracted data within the 2 s window based on the impact and extracted wavelet acceleration signal coefficients as feature vectors from this interval. Five classifiers were compared: ANN (92.2%) < probabilistic principal component analysis (PPCA) (92.2%) < LDA (94.7%) < radial basis function (RBF) (95.0%) < k-NN (97.5%). Previous studies defined specific data sections and extracted discrete-type feature vectors made by compressing multiple frames into one. In addition, ANN exhibited a poor performance compared with other classifiers.

Conversely, some studies reported a good performance in post-fall detection with ANN [19,20]. Yodpijit et al. [19] used one IMU sensor on the waist to distinguish four ADLs and one fall. They used 128 data samples based on the impact and extracted the magnitude of the vectorial sum of acceleration and the magnitude of the vectorial sum of the angular velocity as feature vectors. Their ANN algorithm, fused together with the threshold, resulted in an accuracy of 99.23%. Yoo et al. [20] used one IMU sensor on the wrist to distinguish six ADLs and one fall. All data were unified to have 175 values based on the longest data. The signal magnitude vector (SMV) value of the acceleration and the raw value of acceleration were used as feature vectors. They used only ANN and achieved an accuracy of 100%. Previous studies [19,20] defined a specific data section and extracted time-series feature vectors for the ANN classifier. Even though a good performance was achieved, it depended on the subjects, motions, and classifiers. Therefore, the direct comparison among different studies was relatively difficult.

In general, some data within a dataset were used to develop an algorithm, and the remaining data were used to test the algorithm. However, previous studies analyzed completely different datasets for an effective test, representing here as a cross-dataset. Cao et al. [21] and Delgado-Escaño et al. [22] used different datasets for training and testing to evaluate the algorithm. Cao et al. [21] suggested the adaptive action detection algorithm from human video with high accuracy (95.02%) and used four different datasets to generalize the action detection model. Delgado-Escaño et al. [22] presented a new cross-dataset classifier based on a deep architecture and a k-NN classifier for fall detection and people identification. They tested their algorithm using four different public IMU datasets. Evaluation through a cross-dataset is necessary to apply the algorithm to real situations.

In this study, the performance of algorithms to detect post-fall were evaluated according to classifiers (ANN and SVM) and feature vectors (time-series and discrete data) when untrained motions were used as test data. Some previous studies [19,20], using ANN alone, showed a high accuracy of over 99%, but some [16,17,18], comparing ANN with other traditional classifiers, showed relatively low accuracy. The accuracy of the algorithm depended on the subjects, motions and classifiers, and therefore, a direct comparison among different studies is relatively difficult. SVM was selected as a representative of traditional classifiers to compare with ANN, since it was frequently used in other studies and showed good performance in fall detection. The SisFall dataset [23] was used for cross-dataset. In addition, four different processing conditions (normalization, equalization, increase in the number of training data and additional training with external data) were applied to determine the effect on the performance of the classifiers (ANN and SVM).

## 2. Materials and Methods

### 2.1. Equipment

MPU9250 (InvenSense, San Diego, CA, USA) was used as an IMU, which was placed in the middle of the left and right ASIS (Figure 1a). Three-axis acceleration and three-axis angular velocity were measured at a sampling rate of 100 Hz. The full-scale range of acceleration and angular velocity signals were ±16 g and ±2000°/s, respectively. The data were wirelessly transferred by radio-frequency (RF) communications and stored in synchronization with the video. The GUI was developed using LabVIEW 2019 (National Instruments, Austin, TX, USA) (Figure 1b). Data were analyzed using MATLAB R2020a (MathWorks Inc., Natick, MA, USA).

The SisFall dataset [23] consists of the data measured on the waist with a wearable device, which consisted of two three-axis accelerometers and a three-axis gyroscope sensor. All data were measured at a sampling rate of 200 Hz.

### 2.2. Subjects

Thirty young male volunteers participated in the study (age: 23.6 ± 1.9 years, height: 174.4 ± 5.1 cm, weight: 73.4 ± 8.5 kg). All subjects provided written informed consent before they participated in the study. The study was conducted based on the protocol reviewed and approved by the Yonsei University Research Ethics Committee (1041849-201811-BM-112-01).

The SisFall dataset [23] consists of young adults (11 males; 19–30 years, 1.65–1.83 m, 58–81 kg and 12 females; 19–30 years, 1.49–1.69 m, 42–63 kg) and the elderly.

### 2.3. Experimental Protocol

The experimental protocol consisted of 9 fall motions and 14 ADLs that occurred frequently and were often misidentified (Table 1). Each activity was repeated three times. Only data from young adults were used in this study, since no fall data existed in the elderly group. SisFall dataset [23] consisted of 15 fall motions and 19 ADLs (Table 2).

### 2.4. Preprocessing

The fourth-order Butterworth low-pass filter with a 6 Hz cut-off frequency was used to eliminate high-frequency noise. Putra et al. [24] classified the stages of impact caused by falls: pre-impact, impact and post-impact (Figure 2). Given that the number of samples of the input data had to be matched equally, if the extracted signal contained fewer than standard, the size of the data was maintained with the use of a zero-padding technique.

### 2.5. Feature Vectors

Commonly used feature vectors of short computational time were max, min, mean, variance, skewness and kurtosis with three-directional acceleration and gyro data [16,17]. Figure 3 represents 48 feature vectors for this study.

### 2.6. Window

#### 2.6.1. Sliding Window

Two different sliding windows were used: fixed-size non-overlapping sliding window (FNSW), and fixed-size overlapping sliding window (FOSW) [24], shown in Figure 4a,b, respectively. In this study, the FNSW method was applied to train and test raw data of acceleration and angular velocity. In addition, a sliding window feature (SWF) was applied in which 48 feature vectors were extracted by the FOSW method. The window size was 0.1 s and the overlap was 50%.

#### 2.6.2. Impact-Defined Window

As shown in Figure 5, impact-defined window was determined from the peak value of the SMV of acceleration as the instant of the impact, and then extracted the forward and backward sub-windows based on the impact frame [25]. In this study, the total extracted number of frame was 300, in which backward sub-window size and the forward window size was 1 s and 2 s, respectively. The zero-padding technique was not used even if there were fewer than 300 frames. The impact-defined window feature (IDWF) was applied, in which 48 feature vectors in Figure 3 were extracted from these segmented data.

### 2.7. Classifier

ANN and SVM were used as classifiers. Various classifiers, such as k-NN, NB, LDA and others, were used in previous studies [16,17,18], but all utilized similar clustering-based classifiers. SVM was a simple and powerful classifier, and it had been used in many studies.

#### 2.7.1. ANN

A three-layer ANN was implemented in our study. Two nodes existed in the output layer: fall and ADL. The number of nodes in the input layer was set according to the feature vectors. The number of nodes in the hidden layer was 30. The sigmoid transfer function was used for the activation function. The gradient descent with adaptive learning rate backpropagation was applied.

#### 2.7.2. SVM

Nukala et al. [26] showed that it is more suitable for fall detection to apply the SVM with the RBF kernel rather than SVM with a linear kernel. The other hyperparameters were optimized using the MATLAB toolbox.

### 2.8. Training

The classifiers were trained with five processing conditions:(1)The classifier was trained with the 10 data out of 30 of our laboratory data without any further processing.(2)The min-max normalization was applied in order to reduce biases and variance [16].(3)The data for ADLs were randomly extracted to be equal to that for fall motions [27,28,29,30].(4)The classifier was trained with 20 data rather than 10, out of 30 of our laboratory data.(5)The classifier was trained by adding the data of 13 subjects from the SisFall-dataset.

### 2.9. Test

Two different methods were applied to test performances with different classifiers and feature vectors: internal and external tests. Internal and external tests used the rest of our laboratory data and SisFall dataset, respectively (Figure 6).

For the performance evaluation, the values of sensitivity, specificity and accuracy were calculated as follows:(1)Sensitivity=TP/(TP+FN)×100,
(2)Specificity=TN/(FP+TN)×100,
(3)Accuracy=(TP+TN)/(TP+FP+FN+TN)×100,
where *TP*, *FP*, *TN* and *FN* represent true positive, false positive, true negative and false negative values, respectively.

## 3. Results

Table 3 shows the performance of ANN and SVM classifiers for the internal and external tests according to the feature vectors and different processing conditions. When no processing was used, the following characteristics were shown. For ANN, SWF showed the highest accuracy in both internal and external tests. For SVM, raw data exhibited the highest accuracy in the internal test, and IDWF exhibited the highest accuracy in the external test. It is noted that SWF exhibited a poor performance in SVM, while it generally showed good performance in ANN. Furthermore, both ANN and SVM exhibited good performances when only raw data were used in the internal test. However, the performance was better in the external test when feature vectors were used. For the internal test, normalization processing of feature vectors resulted in decreased performance in ANN, but increased performance in SVM. For the external test in ANN, raw data and IDWF showed increased performance, but SWF showed decreased performance. For the external test in SVM, raw data showed decreased performance, but SWF and IDWF showed increased performance. The equalization processing was not as effective as the normalization. In most cases, the sensitivity increased, but the specificity decreased. When the number of training data increased, for ANN, the performance decreased in the internal test but increased in the external test. For SVM in the internal test, all feature vectors exhibited increased performances. However, for SVM in the external test, the performance of SWF increased, but the performance of raw data and IDWF decreased. The result of additional training with external data was compared with the result of increasing the number of training data. For both ANN and SVM, the overall performances decreased in the internal test but increased in the external test.

False alarms are compared between the increasing a training our laboratory data and additional training of external data, as shown in Table 4 The top two or three false alarms are listed in order. The following major false alarms were detected when the number of training data increased: (a) ADLs with the rapid change in the body COM (YD05,11,06,07, SD05,06,11,13,16,17), (b) fall motions with the slow change in the body COM (YF04,05, SF10,11,12,14,15), (c) lateral lying motions (SD12,13,14) and (d) some lateral falls (SF03,12,15). Table 5 and Table 6 represented the false alarms in the external test with ANN and SWF. Major false alarms for the increase in training data came from significant lateral motions (SD12,13,14 and SF03,12). The additional training with external data reduced those major false alarms significantly. However, some ADLs that involved a rapid change in the body COM (SD13,16) and fall motions that involved a slow change in the body COM (SF11,14) were still falsely detected. 

## 4. Discussion

### 4.1. Feature Vectors

Overall, ANN was more suitable with SWF. Conversely, SVM was more suitable with IDWF. Specifically, for the SWF, the function as a classifier was lost based on the determination of almost all motions as ADL in SVM. However, considering that the function was restored in other processing conditions, it appeared that the information necessary for classification was insufficient. With the exception of this case, SWF and IDWF performed better than Raw for the external test, because raw data of external data have different patterns compared with the existing pattern. However, for the extraction of a specific feature vector from the raw data, it was apparent that the performance of the classifier was guaranteed to some extent because there was a possibility that the feature vector overlapped even with the untrained data. In the previous studies, only IDF [17,18,19,31] and SWF [20,21] were used as feature vectors, respectively, and none of the studies applied all of them to compare. These results suggest that the fall detection performance depends on feature vectors suitable for ANN and SVM.

### 4.2. Normalization

In ANN, the overall performance decreased according to data normalization. The ANN sets the optimal weights and bias for each feature vector for classification according to training. Therefore, there was no need for separate data normalization; instead, it was inferred that the performance decreased because the prominent data patterns were their re-arranged version within the range 0–1. Conversely, in SVM, the overall performance improved according to data normalization. This was thought to be attributed to the fact that SVM was a clustering-based classifier. Given that it was classified based on clustering, the distance between feature vectors became an important factor, and data normalization was effective. In previous studies, the min-max normalization was used as a preprocessing process to detect falls [16,17,32], but there have been no studies comparing the use of normalization. Ozdemir et al. [17] detected between 20 Falls and 16 ADLs using 6 inertial sensors. Similar to this study, ANN (95.68%) showed lower accuracy than SVM (99.48%). Vallabh et al. [16] detected between 4 Falls and 7 ADLs, and SVM (86.75%) showed slightly better performance than ANN (85.87%). Wannenburg et al. [32] classified five ADLs, and ANN (98.88%) showed better performance than SVM (94.32%). When the position of the sensor was fixed and the difference between motions was significant such as falls and ADLs, the normalization process ineffective to ANN. However, the normalization was effective for ANN when classified between similar motions or position of the sensor is fluctuating.

### 4.3. Equalization

The data equalization did not improve significantly to the performance of the classifier, but a more advantageous classifier could be obtained for fall detection. The main problem in imbalanced data is that the majority classes represented by large numbers of patterns rule the classifier decision boundaries at the expense of the minority classes represented by small numbers of patterns [33]. Our own laboratory dataset contained more ADLs than fall motions. Therefore, if the classifier was trained with this dataset, it was trained to become more advantageous in fitting ADLs. However, if the data equalization was used to match the amount of data in each class equally, it was possible to identify more accurately on the fall motions which were the minor class. As a result, the specificity decreased, but the sensitivity increased. In previous studies, the synthetic minority over-sampling technique was used to solve the class imbalancing problem, but there was no study comparing the use of that. Khojasteh et al. [34] tried to detect falls using four public datasets and obtained a sensitivity and specificity of about 90% in both SVM and ANN. Wang et al. [35] used the Sisfall public dataset and obtained a sensitivity and specificity of more than 98% in both SVM and ANN.

### 4.4. Increase in Training Data

The present study showed that the overall classification performance improved, as the number of training data samples increased. As for ANN, the performance improvement was even greater in the external test than in the internal test, since more information was required for the classification in the external test. On the other hand, for SVM, the performance dramatically improved in the internal test, but slightly decreased in the external test. The classification criteria were more clearly determined, as more data samples were used in SVM. However, the overfitting occurred in the external test since untrained data were tested. Kim et al. [36] classified seven hand movements using ANN, as donning and doffing of an EMG sensor-based armband module was repeated. They showed that the classification accuracy increased and the deviation decreased as the number of training data increased.

### 4.5. Additional Training with External Data

The overall performance decreased in the internal test, since unnecessary data were trained. However, the performance increased in the external test, as expected. Based on these results, machine learning was vulnerable to untrained motions, and data on various movements were necessary as training data.

### 4.6. False Alarms

The following major false alarms were detected when the number of training data increased: (a) ADLs with the rapid change in the body COM, (b) fall motions with the slow change in the body COM, (c) lateral lying motions and d) some lateral falls. On the other hand, ADLs with the rapid change in the body COM and fall motions with the slow change in the body COM were falsely detected when the external data was applied for training. The above results could be well understood since lateral fall motions in our laboratory dataset did not include prior motions such slip or trying to sit down, unlike the SisFall dataset. Therefore, various motions should be trained to detect falls based on the machine learning. In addition, additional feature vectors might be needed for the accurate classification.

### 4.7. Limitations

The present study has the following limitations. Only young subjects participated in our lab experiments due to safety issues. In general, the elderly moves less dynamically during ADLs, and thus, the difference between the fall and the ADL signal becomes larger in real situations, which makes it easier to distinguish. In addition, our lab experiments had simulated falls instead of actual ones, which make public fall datasets more important. The more public dataset would be used for the robust fall detection algorithm. It was sufficient to determine the performances of SVM and ANN classifiers, even though only 48 feature vectors were used in this study and an additional methods of feature vector selection, such as the ranking algorithm [37], were not applied.

### 4.8. Future Research

As new tools, such as hardware interfaces, or frameworks, such as Jetson Nano and TensorFlow Lite, were introduced, novel studies were conducted to pre-impact fall detection using deep learning techniques [38,39]. The deep learning techniques, including CNN and LSTM, fit well with sliding window-based feature vectors and were effective to detect pre-impact falls. In future research, deep learning techniques will need to be applied with sliding window-based features to detect pre-impact falls.

## 5. Conclusions

This study showed that the fall detection performance depends on feature vectors for ANN and SVM. Overall, ANN and SVM were suitable for the time-series and discrete data, respectively. When untrained motions were tested using a public dataset, the classification performance generally decreased, and thus, specific feature vectors from the raw data were necessary. Four different processing conditions were applied. Normalization made SVM and ANN more and less effective, respectively. Even though equalization did not improve overall performance, it increased the sensitivity. The increase in the number of training data improved the overall classification performance. Finally, machine learning was vulnerable to untrained motions, and data of various movements were needed for the training.

## Figures and Tables

**Figure 1 sensors-21-04638-f001:**
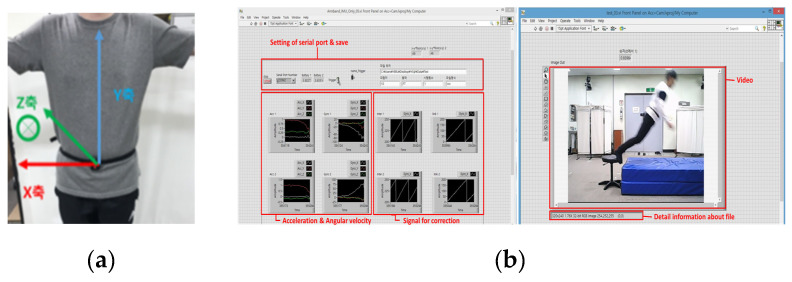
Instruments used in this study: (**a**) IMU and (**b**) GUI.

**Figure 2 sensors-21-04638-f002:**
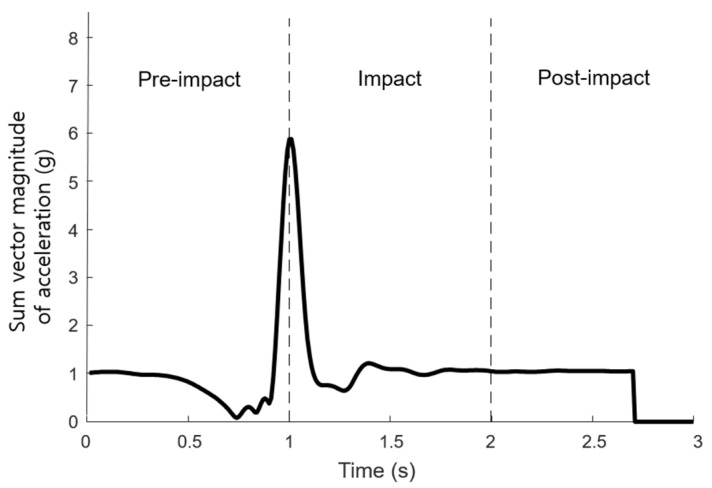
Stage of the impact by falls.

**Figure 3 sensors-21-04638-f003:**
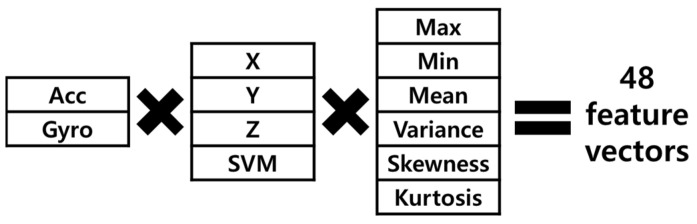
Feature vectors for this study.

**Figure 4 sensors-21-04638-f004:**
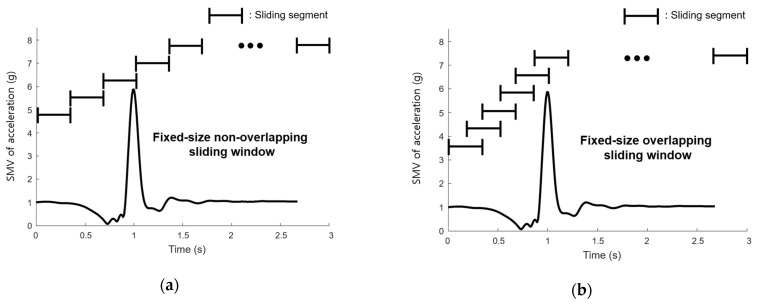
Sliding window: (**a**) FNSW and (**b**) FOSW.

**Figure 5 sensors-21-04638-f005:**
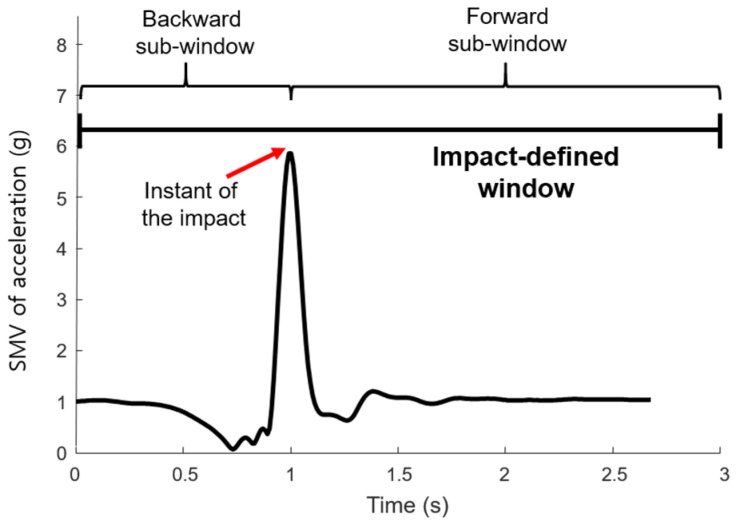
Impact-defined window.

**Figure 6 sensors-21-04638-f006:**
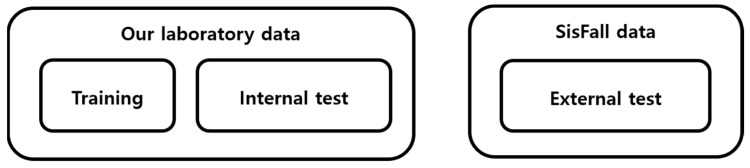
Internal and external tests.

**Table 1 sensors-21-04638-t001:** Fall motions and ADLs used in the study.

Code	Activity
YF01	Slip–backward fall
YF02	Walk-trip–forward fall
YF03	Jogging-trip–forward fall
YF04	Sit down–backward fall
YF05	Sit–backward fall
YF06	Forward fall
YF07	Backward fall
YF08	Lateral fall
YF09	Twist fall
YD01	Walking
YD02	Jogging
YD03	Squat
YD04	Waist bending
YD05	Stumble while walking
YD06	Jogging in place
YD07	Jumping
YD08	Climb stairs up and down
YD09	Slowly sit and stand up from stool
YD10	Quickly sit and stand up from chair
YD11	Collapse in a chair when trying to stand up
YD12	Lying
YD13	Slowly sit and stand up from a low-height mattress
YD14	Quickly sit and stand up from a low-height mattress

**Table 2 sensors-21-04638-t002:** Fall motions and ADLs in SisFall dataset.

Code	Activity
SF01	Fall forward while walking caused by a slip
SF02	Fall backward while walking caused by a slip
SF03	Lateral fall while walking caused by a slip
SF04	Fall forward while walking caused by a trip
SF05	Fall forward while jogging caused by a trip
SF06	Vertical fall while walking caused by fainting
SF07	Fall while walking, with the use of hands on a table to dampen fall, caused by fainting
SF08	Fall forward when trying to get up
SF09	Lateral fall when trying to get up
SF10	Fall forward when trying to sit down
SF11	Fall backward when trying to sit down
SF12	Lateral fall when trying to sit down
SF13	Fall forward while sitting, caused by fainting or falling asleep
SF14	Fall backward while sitting, caused by fainting or falling asleep
SF15	Lateral fall while sitting, caused by fainting or falling asleep
SD01	Walking slowly
SD02	Walking quickly
SD03	Jogging slowly
SD04	Jogging quickly
SD05	Walking upstairs and downstairs slowly
SD06	Walking upstairs and downstairs quickly
SD07	Slowly sit in a half-height chair, wait a moment and stand up slowly
SD08	Quickly sit in a half-height chair, wait a moment and stand up quickly
SD09	Slowly sit in a low-height chair, wait a moment and stand up slowly
SD10	Quickly sit in a low-height chair, wait a moment and stand up quickly
SD11	Sit down for a moment, try to get up and collapse into a chair
SD12	Sit down for a moment, lie down slowly, wait a moment and sit down again
SD13	Sit down for a moment, lie down quickly, wait a moment and sit down again
SD14	While on one’s back, change to a lateral position, wait a moment and change to one’s back
SD15	While standing, slowly bend at the knees and stand up straight
SD16	While standing, slowly bend without bending at the knees and stand up straight
SD17	While standing, get into a car, remain seated and get out of the car
SD18	Stumble while walking
SD19	Gently jump without falling (trying to reach a high object)

**Table 3 sensors-21-04638-t003:** Performance of ANN and SVM.

Processing	Performance	Internal Test						External Test					
		Raw		SWF		IDWF		Raw		SWF		IDWF	
		ANN	SVM	ANN	SVM	ANN	SVM	ANN	SVM	ANN	SVM	ANN	SVM
None	Sensitivity (%)	100.00	99.07	99.81	2.22	99.81	87.04	64.87	62.96	76.99	0.41	70.03	90.72
	Specificity (%)	99.76	96.79	100.00	99.88	99.88	91.19	90.09	95.38	91.41	99.06	95.60	92.85
	Accuracy (%)	99.86	97.68	99.93	61.67	99.86	89.57	77.81	79.59	84.39	51.02	83.15	91.81
Normalization	Sensitivity (%)	100.00	99.63	100.00	100.00	100.00	100.00	94.20	60.46	74.96	70.55	75.07	93.33
	Specificity (%)	92.26	99.88	99.76	99.52	99.52	99.05	73.75	91.74	92.07	91.69	95.05	99.28
	Accuracy (%)	95.29	99.78	99.86	99.71	99.71	99.42	83.71	76.51	83.74	81.39	85.32	96.39
Equalization	Sensitivity (%)	100.00	99.81	99.81	100.00	100.00	93.89	71.19	69.45	78.67	99.48	71.94	93.04
	Specificity (%)	99.76	94.88	100.00	40.83	99.88	88.81	88.88	93.89	91.30	59.93	94.77	90.09
	Accuracy (%)	99.86	96.81	99.93	63.99	99.93	90.80	80.27	81.99	85.15	79.19	83.65	91.53
Increase in training data	Sensitivity (%)	99.63	99.63	100.00	99.26	99.63	95.19	73.57	55.07	78.26	97.57	80.70	89.57
	Specificity (%)	99.76	99.29	100.00	45.71	99.76	96.43	88.11	96.53	91.69	70.50	93.78	93.29
	Accuracy (%)	99.71	99.42	100.00	66.67	99.71	95.94	81.03	76.34	85.15	83.68	87.41	91.47
Additional training with external data	Sensitivity (%)	98.89	98.52	99.81	98.89	99.81	96.48	96.92	96.21	98.56	99.18	99.18	93.33
	Specificity (%)	99.64	92.38	99.52	41.55	98.93	90.95	99.42	94.84	99.61	64.75	100.00	95.33
	Accuracy (%)	99.35	94.78	99.64	63.99	99.28	93.12	98.20	95.50	99.10	81.52	99.60	94.36

**Table 4 sensors-21-04638-t004:** False alarms.

Processing	Classifier		Internal Test			External Test		
			Raw	SWF	IDWF	Raw	SWF	IDWF
Increase in training data	ANN	ADL	YD11	-	YD11	SD14, 12, 13	SD14, 12, 13	SD14, 13
		Fall	YF07	-	YF09	SF12, 15	SF12, 03	SF03, 06, 12
	SVM	ADL	YD11, 05	YD06, 07, 02	YD06, 11	SD13, 16	SD06, 18	SD06, 04
		Fall	YF07	YF02	YF04, 05	SF03, 12	SF14, 10	SF14, 11
Additional training with external data	ANN	ADL	YD04, 11	YD11, 05	YD11, 10	SD16, 13, 07	SD13, 16	-
		Fall	YF05, 07, 08	YF05	YF05	SF11, 14, 15	SF11, 14	SF11, 14
	SVM	ADL	YD06, 11	YD05, 06, 07	YD06, 05, 11	SD06, 18	SD06, 18-	SD06, 17
		Fall	YF07	YF06, 02	YF05, 04	SF14, 15	SF14	SF14, 11

**Table 5 sensors-21-04638-t005:** False alarms of ADLs in the external test with ANN and SWF.

ADLs		**SD01~09,11,15,18,19**	**SD10**	**SD12**	**SD13**	**SD14**	**SD16**	**SD17**
	Increase in training data	0	1	41	41	62	2	4
	Additional training with external data	0	0	0	2	0	2	0

**Table 6 sensors-21-04638-t006:** False alarms of falls in the external test with ANN and SWF.

Falls		**SF01~02**	**SF03**	**SF04**	**SF05**	**SF06**	**SF07**	**SF08**	**SF09**	**SF10**	**SF11**	**SF12**	**SF13**	**SF14**	**SF15**
	Increase in training data	8	43	19	9	24	21	33	32	12	30	58	36	12	30
	Additional training with external data	0	0	0	0	0	1	0	0	0	7	1	0	4	1

## Data Availability

Data available on request due to restrictions e.g., privacy or ethical.

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
