# Peer review of "The Performance of Post-Fall Detection Using the Cross-Dataset: Feature Vectors, Classifiers and Processing Conditions"

_sensors, 2021, doi:10.3390/s21144638_

Round 1

Reviewer 1 Report

The authors propose an evaluation of the performance of different algorithms to detect post-fall. The SisFall dataset was used for cross-dataset evaluation.

This paper presents work in an important field and to the best of my knowledge, the paper is original and unpublished. The paper is well organized.

In the Introduction section, the paper should be devoted to give a comprehensive review of literature, papers on rehabilitation may also be included. It is recommended to analyze better the literature, more recent articles are available:

Evaluation of a Rehabilitation System for the Elderly in a Day Care Center, Information 2019, 10, 3.

In the Discussion section I suggest to add a comparisons with more recent studies, authors used old papers to compare their results.
Moreover, I suggest to exploit Deep Learning and LSTM networks that fit well to sliding window problem.

Author Response

<Response to Reviewer 1 Comments>

 Thanks for your precious comments. The corresponding answer is presented, and some revisions were made based on your comments. (The red-colored sentences were the revised parts.)

Point 1: In the Introduction section, the paper should be devoted to give a comprehensive review of literature, papers on rehabilitation may also be included. It is recommended to analyze better the literature, more recent articles are available:

Evaluation of a Rehabilitation System for the Elderly in a Day Care Center, Information 2019, 10, 3.

Response 1: We revised the manuscript based on your comments. (Line: 38~48)

Many studies have tried to improve the physical performance of the elderly by performing rehabilitation programs to help prevent falls. Røyset et al. [5] conducted a fall prevention program using the Norwegian version of the fall risk assessment method, “STRATIRY” (score 0-5), but achieved no significant improvement when compared to the control group during a short stay in an orthopedic department. Gürler et al. [6] proposed a recurrent fall prevention program including assessment of fall risk factors, education on falls and home modification. This program was effective in reducing fall-related risk factors and increasing fall knowledge. Palestra et al. [7] presented a rehabilitation system based on a customizable exergame protocol (KINOPTIM) to prevent falls in the elderly. As a result of training for 6 months, the performance of the postural response was improved by an average of 80%.

  1. Røyset, B.; Talseth-Palmer, B. A.; Lydersen, S.; Farup, P. G. Effects of a fall prevention program in elderly: a pragmatic observational study in two orthopedic departments. Clinical Interventions in Aging 2019, 14, 145.
  2. Gürler, H.; Bayraktar, N. The effectiveness of a recurrent fall prevention program applied to elderly people undergoing fracture treatment. International Journal of Orthopaedic and Trauma Nursing 2021, 40, 100820.
  3. Palestra, G.; Rebiai, M.; Courtial, E.; Koutsouris, D. Evaluation of a rehabilitation system for the elderly in a day care center. Information 2019, 10(1), 3.

Point 2: In the Discussion section I suggest to add a comparisons with more recent studies, authors used old papers to compare their results.

Response 2:

 Even though the number of previous studies regarding post-fall detection using SVM and ANN is relatively limited, a new reference [31] is added in 4.1. (Line: 275)

4.1. Feature vectors

Overall, ANN was more suitable with SWF. Conversely, SVM was more suitable with IDWF. Specifically, for the SWF, the function as a classifier was astrayed based on the determination of almost all motions as ADL in SVM. However, considering that the function was restored in other processing conditions, it appeared that the information necessary for classification was insufficient. With the exception of this case, SWF and IDWF performed better than Raw for the external test, because raw data of external data have different patterns compared with the existing pattern. However, for the extraction of a specific feature vector from the raw data, it was apparent that the performance of the classifier was guaranteed to some extent because there was a possibility that the feature vector overlapped even with the untrained data. In the previous studies, only IDF [17-19,31] and SWF [20,21] were used as feature vectors, respectively, and none of the studies applied all of them to compare. These results suggest that the fall detection performance depends on feature vectors suitable for ANN and SVM.

  1. Kadhum, A. A.; Al-Libawy, H.; Hussein, E. A. An accurate fall detection system for the elderly people using smartphone inertial sensors. Journal of Physics 2020, 1530(1), 012102.

In addition, we corrected the references. All references except [33] were published between 2014 and 2020.

  1. Vallabh, P.; Malekian, R.; Ye, N.; Bogatinoska, D.C. Fall detection using machine learning algorithms. In 2016 24th International Conference on Software, Telecommunications and Computer Networks 2016, 1-9.
  2. Özdemir, A.T.; Barshan, B. Detecting falls with wearable sensors using machine learning techniques. Sensors 2014, 14(6), 10691–10708.
  3. Gibson, R.M.; Amira, A.; Ramzan, N.; Casaseca-de-la-Higuera, P.; Pervez, Z. Multiple comparator classifier framework for accelerometer-based fall detection and diagnostic. Applied Soft Computing 2016, 39, 94–103.
  4. Yodpijit, N.; Sittiwanchai, T.; Jongprasithporn, M. The development of artificial neural networks (ANN) for falls detection. In 2017 3rd International Conference on Control, Automation and Robotics 2017; 547–550.
  5. Yoo, S.; Oh, D. An artificial neural network–based fall detection. International Journal of Engineering Business Management 2018, 10, 1847979018787905.
  6. Cao, L.; Liu, Z.; Huang, T. S; Cross-dataset action detection. In 2010 IEEE Computer Society Conference on Computer Vision and Pattern Recognition 2010, 1998-2005.
  7. Kadhum, A. A.; Al-Libawy, H.; Hussein, E. A. An accurate fall detection system for the elderly people using smartphone inertial sensors. Journal of Physics 2020, 1530(1), 012102.
  8. Wannenburg, J.; Malekian, R. Physical activity recognition from smartphone accelerometer data for user context awareness sensing. IEEE Transactions on Systems, Man, and Cybernetics: Systems 2016, 47(12), 3142-3149.
  9. Satuluri, N.; Kuppa, M. R. A novel class imbalance learning using intelligent under-sampling. International Journal of Database Theory and Application 2012, 5(3), 25-36.
  10. Khojasteh, S. B.; Villar, J. R.; Chira, C.; González, V. M.; De la Cal, E. Improving fall detection using an on-wrist wearable accelerometer. Sensors 2018, 18(5), 1350.
  11. Wang, G.; Li, Q.; Wang, L.; Zhang, Y.; Liu, Z. Elderly fall detection with an accelerometer using lightweight neural networks. Electronics 2019, 8(11), 1354.
  12. Kim, S.; Kim, J.; Koo, B.; Kim, T.; Jung, H.; Park, S.; Kim, Y. Development of an armband EMG module and a pattern recognition algorithm for the 5-finger myoelectric hand prosthesis. International Journal of Precision Engineering and Manufacturing 2019, 20(11), 1997-2006.
  13. Koo, B.; Kim, J.; Kim, T.; Jung, H.; Nam, Y.; Kim, Y. Post-fall detection using ANN based on ranking algorithms. International Journal of Precision Engineering and Manufacturing 2020, 21(10), 1985–1995.
  14. Musci, M.; De Martini, D.; Blago, N.; Facchinetti, T.; Piastra, M. Online fall detection using recurrent neural networks on smart wearable devices. IEEE Transactions on Emerging Topics in Computing 2020, 3027454.
  15. Yu, X.; Qiu, H.; Xiong, S. A novel hybrid deep neural network to predict pre-impact fall for older people based on wearable inertial sensors. Frontiers in Bioengineering and Biotechnology 2020, 8, 63.

Point 3: Moreover, I suggest to exploit Deep Learning and LSTM networks that fit well to sliding window problem.

Response 3:

This study focused on ANN and SVM. The following sentence were added in the introduction section.
(Line: 115~121)

Some previous studies [19,20] using ANN alone showed a high accuracy of over 99%, but some [16-18] comparing ANN with other traditional classifiers showed relatively low accuracy. The accuracy of the algorithm depended on the subjects, motions and classifiers, and therefore the direct comparison among different studies is relatively difficult. SVM was selected as a representative of traditional classifiers to compare with ANN, since it was frequently used in other studies and showed good performances in fall detection.

Thank you for letting us know that deep learning techniques including LSTM are suitable for sliding windows. These deep learning techniques will be applied in future studies.

We also added the following sentences based on your comments. (Line: 353~359)

4.8. Future study

As new tools such as hardware interfaces or frameworks like Jetson Nano and TensorFlow Lite were introduced, novel studies were conducted to pre-impact fall detection using deep learning techniques [38,39]. The deep learning techniques including CNN and LSTM fit well to sliding window-based feature vectors and were effective to detect pre-impact falls. In the future study, deep learning techniques will be applied with sliding window-based features to detect pre-impact falls.

  1. Musci, M.; De Martini, D.; Blago, N.; Facchinetti, T.; Piastra, M. Online fall detection using recurrent neural networks on smart wearable devices. IEEE Transactions on Emerging Topics in Computing 2020, 3027454.
  2. Yu, X.; Qiu, H.; Xiong, S. A novel hybrid deep neural network to predict pre-impact fall for older people based on wearable inertial sensors. Frontiers in Bioengineering and Biotechnology 2020, 8, 63.

Reviewer 2 Report

This is a well-written paper. We recommend to accept this paper with the following minor modifications:

  1. Please state more clearly why selecting SVM and ANN as the classifiers to be compared in this paper.
  2. The comparison results are mainly shown in Table 3 and Table 4. Is it possible to provide more figures to visualize the insights of these performance data ?

Author Response

<Response to Reviewer 2 Comments>

Thanks for your precious comments. The corresponding answer is presented, and some revisions were made based on your comments. (The red-colored sentences were the revised parts.)

Point 1: Please state more clearly why selecting SVM and ANN as the classifiers to be compared in this paper.

Response 1: We added the following sentences to the manuscript based on your comments. (Line: 115~121)

Some previous studies [19,20] using ANN alone showed a high accuracy of over 99%, but some [16-18] comparing ANN with other traditional classifiers showed relatively low accuracy. The accuracy of the algorithm depended on the subjects, motions and classifiers, and therefore the direct comparison among different studies is relatively difficult. SVM was selected as a representative of traditional classifiers to compare with ANN, since it was frequently used in other studies and showed good performances in fall detection.

Point 2: The comparison results are mainly shown in Table 3 and Table 4. Is it possible to provide more figures to visualize the insights of these performance data?

Response 2:

In the case of Table 3, we thought a lot to visualize it using another way, but it seems to be the best way to express our data since it contains too much data.

In the case of Table 4, we added Table 5 to explain more detail based on your comments. (Line: 254~259) Please see the attached Word file for Table 5.

Table 5 represents the false alarms in external test with ANN and SWF. Major false alarms for the increase in training data came from significant lateral motions (SD12,13,14). The additional training with external data reduced those major false alarms significantly. However, some ADLs with the rapid change in the body COM (SD13,16) and fall motions with the slow change in the body COM (SF11,14) were still falsely detected.

We also added more detailed explanation about Table 4. (Line: 249~254)

The top two or three false alarms are listed in order. The following major false alarms were detected when the number of training data increased: a) ADLs with the rapid change in the body COM (YD05,11,06,07, SD05,06,11,13,16,17), b) fall motions with the slow change in the body COM (YF04,05, SF10,11,12,14,15), c) lateral lying motions(SD12,13,14), and d) some lateral falls(SF03,12,15).
